complexity/mathematical modelling

vaccination, evolutionary game theory, cooperation, social dilemma

**Author for correspondence:**
Matjaz Perc
e-mail: matjaz.perc@gmail.com

# Optimal governance and implementation of vaccination programmes to contain the COVID-19 pandemic

Mahendra Piraveenan[1,2], Shailendra Sawleshwarkar[3,5,8], Michael Walsh[4,5], Iryna Zablotska[3,5], Samit Bhattacharyya[9], Habib Hassan Farooqui[8], Tarun Bhatnagar[10], Anup Karan[8], Manoj Murhekar[10], Sanjay Zodpey[8], K. S. Mallikarjuna Rao[11], Philippa Pattison[7], Albert Zomaya[6] and Matjaz Perc[12,13,14]

[1]Complex Systems Research Group, Faculty of Engineering, [2]Charles Perkins Centre, [3]Westmead Clinical School, Faculty of Medicine and Health, [4]School of Public Health, Faculty of Medicine and Health, [5]Marie Bashir Institute of Infectious Diseases and Biosecurity, [6]School of Computer Science, Faculty of Engineering, and [7]Office of the Deputy Vice-Chancellor, University of Sydney, New South Wales 2006, Australia
[8]Public Health Foundation of India, Delhi, India
[9]Department of Mathematics, School of Natural Sciences, Shiv Nadar University, Uttar Pradesh, India
[10]ICMR-National Institute of Epidemiology, Chennai, India
[11]Industrial Engineering and Operations Research, Indian Institute of Technology Bombay, Mumbai, India
[12]Faculty of Natural Sciences and Mathematics, University of Maribor, Maribor, Slovenia
[13]Department of Medical Research, China Medical University Hospital, China Medical University, Taichung, Taiwan
[14]Complexity Science Hub Vienna, Vienna, Austria

MP, 0000-0001-7073-1421; SS, 0000-0002-9399-0490; SB, 0000-0002-7357-7949; MP, 0000-0002-3087-541X

Since the recent introduction of several viable vaccines for SARS-CoV-2, vaccination uptake has become the key factor that will determine our success in containing the COVID-19 pandemic. We argue that game theory and social network models should be used to guide decisions pertaining to vaccination programmes for the best possible results. In the months following the introduction of vaccines, their availability and the human resources needed to run the vaccination programmes have been scarce in many countries. Vaccine hesitancy is also being encountered from some sections of the general public.

We emphasize that decision-making under uncertainty and imperfect information, and with only conditionally optimal outcomes, is a unique forte of established game-theoretic modelling. Therefore, we can use this approach to obtain the best framework for modelling and simulating vaccination prioritization and uptake that will be readily available to inform important policy decisions for the optimal control of the COVID-19 pandemic.

# 1. Introduction

Vaccination is the main hope to contain the COVID-19 pandemic currently enveloping the world [1,2]. Several vaccines for SARS-CoV-2 have been developed or are under development, and their potential impact and limitations are hotly debated. It is expected that herd immunity [3–6] will play a role in containing the pandemic once a sufficiently high proportion of the world population gains adaptive immunity. Nevertheless, if high morbidity, mortality and economic catastrophe are to be avoided, the vast majority of the population should acquire immunity through vaccination rather than infection. While global eradication of COVID-19 could be a difficult goal to achieve [7,8], a successful vaccination programme may target regional elimination in the short to medium term. Hence, vaccination uptake will have a direct and critical influence on the dynamics of the COVID-19 pandemic, and the ability of the healthcare systems to manage it.

In this position paper, we argue that the level of vaccination uptake in populations, the effective prioritization of potential vaccine recipients and the efficient use of the resources needed by vaccination administration programmes will be the key determinants in how the COVID-19 pandemic is contained and/or eliminated in populations in the coming years. We posit that without widespread uptake, any vaccination programme will fail regardless of the efficacy [9,10] of the vaccine itself. Indeed, there is some evidence to suggest that higher efficacy of vaccines, which is desirable in itself, decreases vaccination uptake by encouraging free-riding behaviour [11]. Therefore, what we describe in this position paper are conditions under which the vaccination programmes can achieve their maximum uptake and effectiveness, and the modelling and implementation approaches needed to help the vaccination programmes achieve this.

We propose that game-theoretic modelling, which is the established theoretical framework for modelling rational decision-making [12,13], coupled with social network analysis and agent-based techniques for modelling the population and simulating the disease dynamics [4,14–18], will give us the most effective toolset to model the vaccination uptake. This multi-faceted approach will produce a definitive roadmap for the implementation of vaccination programmes which will lead to successful long-term management of COVID-19.

## 1.1. Position statements

**1.1** In the coming years, the most important factors that will determine the success of controlling the COVID-19 pandemic will be the level of vaccination uptake by the population, and the effective use of resources to administer the vaccine in highly varied and variable settings.

**1.2** Game theory, supplemented by social network analysis and agent-based modelling, should be extensively used by researchers to model vaccination uptake by populations and guide difficult policy decisions regarding vaccination programmes and thus maximize containment of the COVID-19 pandemic.

**2.1** Given the limited availability of vaccines at the initial stages, effective prioritization and optimal use of resources are crucial. If the vaccine is of the type which reduces the transmissibility of the SARS-CoV-2 virus, prioritization should be based on targeting individuals, cities or states that can act as critical nodes of transmission or superspreaders. Whereas if the vaccine is of the type that reduces symptoms or mortality, prioritization should be based on targeting individuals, cities or states that are likely to have poor outcomes if infected.

**2.2** Game theory, together with social networks and agent-based modelling, can be used as a primary theoretical framework in determining effective prioritization of scarce resources needed in different vaccination programmes, depending on a broad range of bounding conditions for successful implementation.

# 2. Background

## 2.1. The current state of SARS-CoV-2 vaccine development and delivery

Effective SARS-CoV-2 vaccine development, production and dissemination are expected to take at least 12–18 months [19,20]. Efforts to develop vaccines have explored different approaches, ranging from recombinant vaccines and nucleotide-based vaccines to subunit vaccines and others. Most vaccines currently being administered are aimed at inducing neutralizing antibodies against the viral spike (S) protein of SARS-CoV-2, which prevent binding to ACE2 receptors of host cells [2]. The leading vaccines currently being administered are (i) the ChAdOx1 vaccine developed by the University of Oxford and AstraZeneca, (ii) the Pfizer-BioNTech BNT162 vaccine, (iii) the Moderna RNA vaccine, which is designed to induce antibodies against a portion of the coronavirus S protein, (iv) the Sinopharm inactivated SARS-CoV-2 vaccine, (v) the Gameliya Research Institute Gam-COVID-Vac Adeno-based vaccine, and (vi) Bharat Biotech whole-virion inactivated SARS-CoV-2 vaccine (BBV152). Currently, WHO lists another 100 vaccine candidates as under investigation in human clinical trials and 184 in preclinical trials [21]. Out of these, 19 vaccine candidates are in phase 3 clinical trials at present, with 16 other candidates in phase 2 or phase 2/3.

The global collaborative effort, COVAX, has been initiated to negotiate and ensure equitable access of SARS-CoV-2 vaccines to all participating countries regardless of income levels [22]. Several countries have also made alternative arrangements for procuring approved vaccines directly from the manufacturers. Nevertheless, despite unprecedented levels of accelerated research and international cooperation, most countries are finding it difficult to procure sufficient doses to vaccinate their entire populations, and may not get sufficient vaccine supplies to do so in the near future.

## 2.2. SARS-CoV-2 vaccination objectives and the role of herd immunity

Achieving herd immunity is often a key objective of population-level vaccination coverage. Herd immunity [23] is a population threshold that marks the necessary proportion of the population that needs to be immune to an infection, either through vaccination or through exposure to the pathogen, so that the transmission of the infectious agent is sufficiently disrupted, and the entire population is protected [3]. It is not desirable to expose a significant portion of the population to the pathogen in order to acquire herd immunity. Rather, the objective should be to achieve herd immunity through vaccination to minimize morbidity and mortality.

The level of herd immunity needed to protect a population can be derived from the basic reproduction number ($R_0$), which is defined as the number of secondary infections produced on average by an infected index case within a completely susceptible population, assuming there is no human intervention [4–6]. The epidemic threshold is defined as the inverse of the basic reproduction number [5,6]. The level of herd immunity needed to protect a population from an epidemic is equal to the complement of the epidemic threshold (i.e. herd immunity threshold $= 1 - 1/R_0$).

The basic reproduction number $R_0$ for COVID-19 and the corresponding herd immunity threshold are not yet known definitively, with current best estimates of $R_0$ ranging from 2.5 to 3.0 [24–28], with the corresponding herd immunity thresholds ranging from 60 to 67%. Furthermore, much uncertainty remains regarding the nature of the immunogenicity of the pathogen, with important implications for vaccines. It is currently unknown whether humoral or cell-mediated responses drive neutralizing immunity or other correlates of protection, and how long any such protection endures [29–31]. If vaccination does not generate protective immunity in every member of the population who gets vaccinated, then the number of people requiring vaccination for the population to achieve herd immunity will be higher than the number determined by the herd immunity threshold as defined by $R_0$. Similarly, waning vaccine-induced immunity will require greater population coverage than that derived from $R_0$, and also may necessitate the administration of booster vaccination. Also, the targeting of epidemiologically influential subgroups will be important, and the relative importance of some of these subgroups is disease-specific. For example, healthcare and other essential workers may be important subgroups to target for SARS-CoV-2 vaccination due to their relatively high levels of exposure. According to recent evidence [32,33], children also may be more influential to transmission than previously assumed. Therefore, despite the transient immunogenicity of SARS-CoV-2, targeted vaccination delivery may achieve herd immunity at or below the threshold derived from $R_0$ if epidemiologically influential subgroups are prioritized. All these

aspects of herd immunity specific to COVID-19 will necessarily be foundational to the modelling of SARS-CoV-2 vaccine effectiveness.

## 2.3. Game theory in vaccination uptake modelling

Game theory, which is the study of strategic decision-making by rational players, is used to study several phenomena and behavioural patterns in human societies and socio-economical systems, and is applied in fields ranging from evolutionary biology to computer science and project management [4,34–38]. Several previous studies have modelled vaccination uptake using game theory in the context of diseases such as influenza, measles, chickenpox and hepatitis [4]. When modelling vaccination uptake using game theory, *players* usually represent individuals, and the actions involve taking or not taking a vaccine. The *payoff* is decided by several factors including the perceived risk of infection (perceived prevalence and transmissibility), severity of the disease, financial and non-financial cost of vaccination, and the perceived uptake of vaccination by other players.

To reach a decision, individuals either try achieving utility maximization or comparing the payoffs of two strategies. The decision-making process is modelled in two ways: (i) self-learning (*Aspiration game*) [39] and (ii) social learning (*Imitation game*) [4,40]. Through self-learning, individuals rely on their knowledge, memory and personal perception, and awareness of the disease, and switch strategies if their own aspiration level is not met, while imitation dynamics put individuals into an environment where personal decisions are influenced by the choices of the population, and individuals update their strategies by comparing their own expected payoffs with others in the population and switch to the strategy which gives the better payoff [39–41]. The imitation game can be played in a homogeneous population structure representing a well-mixed population, or a heterogeneous population structure, where the influence by neighbours is determined by the number and strength of contacts between the individual and his/her neighbours [42,43]. Recent research has suggested that imitation dynamics is typically insufficient to sustain herd immunity of a society in the long term [43].

# 3. Vaccination uptake as a key determinant of COVID-19 containment success

## 3.1. Key drivers and barriers of vaccination programmes

Levine [44] describes six key drivers that may influence vaccine uptake. These include the epidemic potential of the pathogen, localized transmission potential of the pathogen, safety concerns with the vaccine, strength and flexibility of public health delivery systems, public investment in resources for immunization, and local ownership and individual normative behaviours. The level of vaccine hesitancy by the population [45], and the local context and its multifactorial determinants also need to be considered.

Assuming that safe and effective SARS-CoV-2 vaccines will be available after timely regulatory approvals, countries will still require enormous resources and systems in place to address vaccination programme implementation challenges. The challenges in vaccination programme implementation include vaccine procurement and supply chain management, developing and deploying vaccine delivery platforms, developing vaccine delivery strategies including identification of eligible/target subpopulations for vaccination, training of frontline workers and social mobilization [1]. These barriers and challenges must also be viewed in the light of the shortages of health workers that exist in many parts of the world. The High-Level Commission on Health, Employment and Economic Growth of WHO [46] noted in 2016 that there was a global shortage of 180 million health workers to meet the prescribed minimum threshold of 44.5 health workers per 10 000 individuals. Given these pre-COVID-19 deficits, the optimal management of human resources to meet the increased demands of the pandemic, manage the normal caseloads from other diseases and still allocate sufficient human resources for the urgent task of vaccination will be a significant challenge to many countries.

## 3.2. Comparative significance of vaccination uptake

We posit that strategic vaccine delivery and its uptake comprise the most important determinants of containing COVID-19 successfully, and compare favourably against considerations related to vaccine efficacy. Vaccine efficacy is defined as the percentage reduction of disease in a vaccinated group of

people compared to an unvaccinated group, using the most favourable conditions [9,10]. Modelling suggests that a vaccine with partial efficacy may have a significant impact and cost-effectiveness with maximal gains achieved with the earlier introduction [1]. Therefore, we argue that any vaccine candidate which has met the minimum endpoints explicitly defined in their phase 3 protocols must be deployed immediately, and the efficacy of such a vaccine, while important, is a concern secondary to achieving strategically targeted coverage as early as possible.

Similarly, we argue that while non-pharmaceutical interventions (NPIs) will retain their significance in containment efforts in the short to medium term, vaccination uptake will eventually supplant them as the primary means to containing COVID-19. Several countries have used contact tracing and isolation efforts, coupled with mask-wearing and social distancing measures, to contain COVID-19, as an interim measure until mass vaccination programmes are successfully implemented [47]. When implemented consistently, these efforts have enjoyed considerable success, but are also both highly resource-intensive and socio-economically disruptive [47,48]. As such, they may be unsustainable over the long term due to population fatigue, economic hardship and the lack of requisite public health resource capital [49]. Therefore, while they are crucial to the containment of COVID-19 in the short term, the importance of vaccination efforts will surpass them in significance in the long term.

# 4. Applying game theory in SARS-CoV-2 vaccination uptake modelling

## 4.1. Why game theory?

There are many compelling reasons why game theory should be employed in the modelling and analysis of SARS-CoV-2 vaccine uptake. Firstly, compulsory vaccination is likely to encounter a level of resistance from the public, and there may not be enough vaccines to vaccinate everyone. Therefore, decisions will have to be made by governments and policymakers about who to vaccinate, and by individuals about whether they want to take the vaccine. An individual is likely to get vaccinated only if the policymakers decide to offer vaccination to that individual, and the individual decides to accept it. Game theory can be used to explicitly model the decision-making of policymakers (vaccine givers) and individuals (vaccine takers), because it has well-defined branches which model 'public good' decision-making (cooperative game theory) and selfish decision-making (non-cooperative game theory) [12,37].

Secondly, game theory can also model the evolution of strategies. In the COVID-19 context, strategies will evolve as more information becomes available about the disease itself and about the vaccines. Different strategies will contest for primacy, and evolutionary game theory [50–52] can explicitly model this.

Thirdly, game theory can explicitly account for 'bounded rationality'—the limited ability of people to assess reality. Thus, game theory can model the perceived and real payoffs of vaccination and distinguish between them, which is important with respect to COVID-19 because of the amount of misinformation and conspiracy theories present.

Finally, game theory can be used in conjunction with other tools, such as prospect theory [53], Monte Carlo simulation [54] and agent-based models [55], which will be useful in modelling the COVID-19 dynamics, and computing and predicting epidemic parameters which can then be used as input to model decision-making, completing the feedback loop. It can also employ supercomputing resources in calculating equilibrium solutions [56,57].

## 4.2. Factors and parameters

A complex array of factors, parameters, drivers and attributes, which influence the decision by individuals to take the vaccine, the decision by governments and policymakers to provide a vaccine to certain people, or both, need to be considered in modelling SARS-CoV-2 vaccination uptake. The age and gender distributions of people will influence vaccination uptake, since older people are more likely to be adversely affected by COVID-19, and likely to have higher mortality rates [24,58], and thus may have more incentive to take the vaccination. Similarly, some preliminary studies suggest that [59,60] men compared to women are more likely to be symptomatic with COVID-19 or to have higher morbidity, thus men may have comparatively more incentive to take vaccination. The cost and accessibility of the vaccine will obviously influence the levels of uptake. In some countries, governments may make the vaccination compulsory for certain demographics, such as people above a

certain age, and this may reduce the incentive for others to vaccinate voluntarily. Vaccination cost may influence uptake in a non-trivial manner, with some studies suggesting that hysteresis loops of vaccination uptake levels may occur with respect to changes in the perceived vaccination cost as well as the vaccine efficacy [61].

The type of vaccine will influence the uptake. We need to model scenarios where the vaccine will be one-off, seasonal or a chemoprophylaxis. We also need to consider whether the vaccine will reduce transmissibility or will reduce the severity of the symptoms and the mortality rate. If the vaccine will reduce transmissibility, then relatively young people, who are more likely to travel and interact with others, should be given preference by policymakers in vaccine access. On the other hand, if the vaccine primarily reduces symptoms or mortality, then older people, and people who have illnesses that increase the likelihood of a poor outcome, will have a higher incentive to take the vaccine, and policymakers will have a higher incentive to give the vaccine to them. The type of vaccine may also affect the evolution of pathogen virulence, and this is an important consideration in the context of new and highly virulent strains of SARS-CoV-2 emerging recently. Some studies argue that vaccines designed to reduce pathogen growth rate and/or toxicity may result in the evolution of more virulent pathogens, thus diminishing the benefits of mass vaccination, while vaccines that reduce transmission do not produce this effect [62,63]. The evolution of pathogen virulence affects the disease dynamics which will in turn influence patterns of vaccine uptake.

Other influential factors are epidemiological metrics, such as COVID-19 incidence, prevalence and cumulative incidence, and these need to be considered at suburb, city, state and country levels, creating a complex array of parameters. Furthermore, the perceived epidemiological parameters and perceived risks of vaccination [64] can differ from real parameters and real risks of vaccination if misinformation is being spread, and this difference between perceived and real parameters can be correlated to the 'bounded rationality' [13,65–67] of the potential vaccinees, or the level of 'noise' present in the information. All such context and nuance will need to be modelled.

The level of interaction a person expects to have with the community in general, and other SARS-CoV-2-infected people in particular, will influence vaccination decisions, and will have to be modelled. For example, a person who travels to work by train, or is employed in a people-facing job such as teaching or food service work, may be more likely to vaccinate compared to a person working from home. Similarly, health workers are more likely to vaccinate. The social structure or topology of a person's immediate neighbours also may influence vaccination decisions [68].

Other factors involve the health of the potential vaccinees. In particular, the perceived and real levels of immunity of people have to be modelled, because people who have relatively higher levels of immunity have less incentive to take vaccination. Similarly, the overall health of the vaccinee, including the presence of chronic diseases, such as hypertension, diabetes and chronic heart diseases, and the perceived correlation of these conditions with severe COVID-19 [69,70] will need to be modelled, as will the perceived and real likelihood of adverse side effects from the vaccine (especially in individuals with other chronic diseases).

Finally, the logistic and human resource management challenges in distributing and administering the vaccine, such as transport of vaccine, storage of vaccine, availability of clinical and support staff to administer the vaccine, and the durability of vaccine after manufacture are factors that need to be considered and modelled in understanding the SARS-CoV-2 vaccination uptake.

## 4.3. Coupling between game theory and simulation

Usually, when the prevalence of a disease decreases, the perceived risk decreases, resulting in a smaller perceived payoff for taking the vaccine, which will in turn result in less vaccine uptake. This may over time result in the re-emergence of disease, which will in turn increase the payoff, so more people will take the vaccine. Therefore, there is a risk that COVID-19 will become endemic, and go through endemic stable cycles (and vaccine uptake will also go through corresponding cycles), if the perceived 'payoff' for taking the vaccine is not high enough on average when the prevalence becomes relatively low [4]. It will be important to establish the conditions under which the SARS-CoV-2 vaccination uptake will go through such cycles, in different countries and subdemographics, so that policy decisions can be made that ensure the payoff for the vaccine is always high enough to avoid such endemic stable cycles. As such, game-theoretic modelling of vaccination uptake has to be tightly coupled with high-fidelity simulation modelling [14,15] of population demographics and disease dynamics, using techniques such as agent-based modelling, to gain a holistic picture about vaccination uptake.

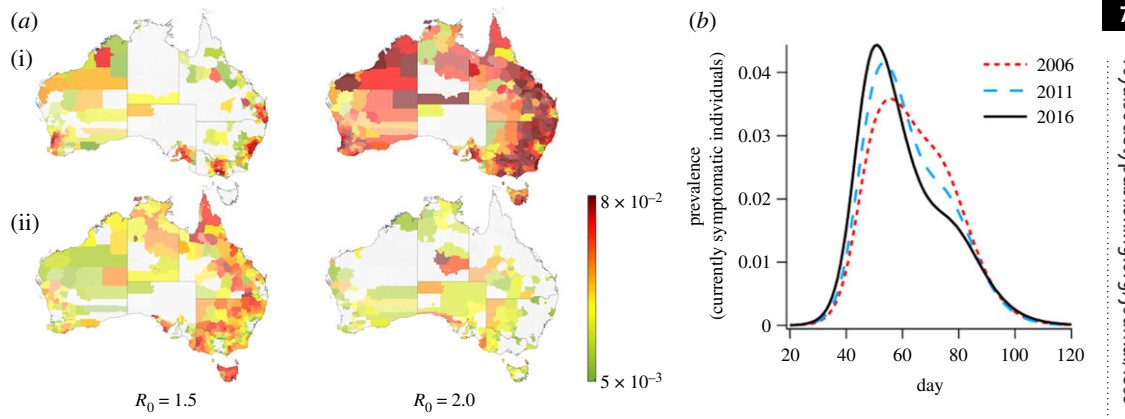

**Figure 1.** Simulation of epidemic spread in Australia using the ACEMod platform ((*a*) adapted from [15]—reprinted with permission from Elsevier; (*b*) adapted from [14]—reprinted with permission from exclusive licensee American Association for the Advancement of Science). (*a*) Prevalence proportion choropleths showing the spatial distribution of simulated epidemics in Australia for $R_0 = 1.5$ and $R_0 = 2.0$. The minimum prevalence (green) is $5 \times 10^{-3}$ and the maximum prevalence (red) is $8 \times 10^{-2}$. The distribution is shown for days 62 (i) and 88 (ii). Both simulations are sample realizations comprising the same demographics (contact) and mobility networks, as well as identical seeding at the same rate at major international airports around Australia. The epidemic peaks at larger cities at similar times, whereas less populous areas are less likely to synchronize [14]. (*b*) The ensemble average of prevalence for simulated influenza epidemics in 2006, 2011 and 2016, with clear trends in the increased peak prevalence and faster spreading rates [15].

In particular, large-scale agent-based simulation methods [55] can be used to model population demographics, including age, gender and household distributions of people, commuting and travel patterns, workplace locations and size distributions, school location and size distributions, and the mixing patterns of people [15] (figure 1). These can in turn be used to estimate epidemic parameters such as transmissibility, basic reproduction number, effective reproduction number, incidence, prevalence and cumulative incidence. An inter-city flux model [4,16–18], populated from census data [14,15], could be used to model travelling patterns and their effect on infection spread.

# 5. Applying game theory in the allocation and prioritization of resources

The allocation, prioritization and distribution of scarce resources needed for implementation is an important aspect of any vaccination programme, particularly in countries like India and Brazil with large populations and moderate *per capita* spending on health [71]. Such resources include human resources, vaccine resources, rolling stock and storage facilities. As a result, targeting decisions inevitably will need to be made [72,73]. Therefore, the vaccination programme becomes a resource allocation problem, and modelling optimal vaccination resource allocation is essential.

In scenarios where limited resources have to be optimally distributed and used, cooperative game theory can be applied with maximum benefit. Often, the outcome of a cooperative game played in a system is equivalent to the result of a constrained optimization process [74]: therefore, such cooperative games are often solved by using a linear programming framework or other optimization tools. In the context of SARS-CoV-2 vaccination, minimization of epidemic parameters, such as incidence or prevalence, minimization of economic costs of the pandemic, minimization of mortality and minimization of disruption to daily lives could be some of the goals of governments and policymakers in deploying vaccines. Therefore, vaccination prioritization could be modelled as a constrained multi-objective optimization problem in real time [75], and cooperative game theory, again coupled with simulation modelling techniques, could be used to solve it.

Clearly, the multi-objective optimization will have to take into account the nature of the vaccine: if the vaccine reduces the transmissibility of SARS-CoV-2, prioritization should be given to targeting individuals, cities or states that can act as critical nodes of transmission or superspreaders. Therefore, the goal will be to reduce transmission, or the number of diagnosed cases. Whereas if the vaccine reduces symptom severity or mortality, prioritization should be given to individuals, cities or states that are likely to have poor outcomes. Therefore, the goal will be to reduce mortality and morbidity.

# 6. Conclusion

In this position paper, we articulated and discussed the view that vaccination uptake will have the most significant influence in the ultimate control of the COVID-19 pandemic, and effective modelling of uptake using appropriate tools, therefore, is of paramount importance. We presented the case for game theory, coupled with simulation techniques, social network analysis and agent-based modelling, to be the most significant mathematical and computational toolset available for this modelling. We highlighted that while the efficacy of the vaccines developed, as well as the efficiency of testing, contact tracing and isolation procedures, shall remain important factors in containing the COVID-19 pandemic, the effectiveness of the vaccination programme and the level of vaccination uptake will surpass these factors in significance in the effort to finally contain, locally eliminate and globally stabilize the COVID-19 pandemic.

We discussed the modelling of vaccination decision-making in detail, and articulated that this has two components: (i) the decision-making process by individuals to get the vaccine and (ii) the decision-making process by governments and policymakers to choose vaccinees, given the reality of limited vaccine doses at the initial stages of vaccination. We argued that the individual decision-making regarding vaccination uptake is influenced by a range of factors including demographics, physical location, level of interaction, the health of the vaccinee, epidemic parameters and perceptions about the vaccine being introduced. Similarly, the decision-making of the government will be influenced by epidemic parameters, the nature of the vaccine being introduced, logistics, management of human resources needed for the vaccination effort and the amount of vaccine doses available. We explained that non-cooperative game theory is ideally suited for modelling individual decision-making behaviour regarding vaccination, while cooperative game theory can be used to inform government decisions regarding prioritization.

The suggested approach is not without challenges. In particular, it is clear that a large array of factors influence vaccination uptake, and capturing them all in the form of utility functions used in game theory will be particularly challenging. At present, we have limited understanding about the rapidly evolving SARS-CoV-2 pathogen, and some initial modelling may soon become outdated as the pathogen evolves further and new strains emerge. Modelling efficacy of different vaccines against the new strains might be particularly challenging, as their efficacy was initially measured against strains which were spreading at the time of clinical trials. Despite these challenges, we believe that this position paper provides a comprehensive roadmap for modelling vaccination uptake, and will stimulate research and deliberation among all stakeholders which will aid the successful implementation of vaccination programmes against COVID-19 and its decisive containment soon.

Data accessibility. We would like to state that there are no data, code or other relevant material associated with the paper.
Authors' contributions. All authors have contributed equally to design, research and writing.
Competing interests. At the time of writing, M.P. is an Editorial Board Member of Royal Society Open Science, but had no involvement in the review or assessment of the paper.
Funding. M.P. was supported by the Slovenian Research Agency (grant nos. P1-0403, J1-2457, J4-9302 and J1-9112).

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
