## [Peer Review File · Royal Society Open Science]

Review History

RSOS-210429.R0 (Original submission)

Review form: Reviewer 1 (Marco Alberto Javarone)

Is the manuscript scientifically sound in its present form?

Yes

Are the interpretations and conclusions justified by the results?

Yes

Is the language acceptable?

Yes

Do you have any ethical concerns with this paper?

No

Have you any concerns about statistical analyses in this paper?

No

Recommendation?

Accept as is

Comments to the Author(s)

In this work, authors study a very important and modern topic, i.e. strategies for containing SARS-COV-2 in the light of the production of vaccines, now available in some countries. Notably, they argue that the optimal strategy for distributing vaccines should result from an approach able to exploit both game theory and social network analysis.

The introduction reads very well, it is compact and presents the essential elements to the reader. The position statements are clear, as the authors clarify that their statements are based on a list of assumptions they explain. The background material is very well-written, also for non-expert in the field, providing all important definitions, as the value R_0 .

The motivation for using Game Theory is given in Section 2.3 and 4.1.

Then in 4.3 it is discussed how to couple this approach with simulations. Finally, Conclusion section provides a general overview of this contribution, highlighting motivations, methods, and so explaining why this strategy should be actually considered/implemented by various governments.

In my opinion, as I above mentioned, this paper is very well-written and it covers a modern aspects in the current pandemic scenario. The discussion authors propose deserve full attention, not only from the scientific community. Their analysis is supported by clear statements and assumptions. For these reasons, I am glad to recommend this paper in the current form.

As optional comment, I would like to mention two investigations that authors could consider when preparing the final version of this work: 'An epidemiological model with voluntary quarantine strategies governed by evolutionary game dynamics, Chaos, Solitons & Fractals 143, 110616, 2021' where a model based on an Evolutionary Game is integrated in the dynamics of a SIR model, and "Heterogeneity in evolutionary games: an analysis of the risk perception, Proceedings of the Royal Society A 476 (2237), 20200116, 2020".

Notably, the first manuscript help to corroborate the the main claim proposed here, on the relevance of using game theory-based approach for dealing with COVID19 and similar scenarios. The second paper can be a useful reference for those readers interested in the impact of risk perception in dilemma games.

To conclude, I wish to congratulate with the authors for a such interesting and clear work.

Review form: Reviewer 2**Is the manuscript scientifically sound in its present form?**

Yes

Are the interpretations and conclusions justified by the results?

Yes

Is the language acceptable?

Yes

Do you have any ethical concerns with this paper?

No

Have you any concerns about statistical analyses in this paper?

No

Recommendation?

Accept with minor revision (please list in comments)

Comments to the Author(s)

In their paper authors study a critically relevant and important problem that can help better stem the burden of the COVID-19 pandemic with proper vaccination governance and implementation. In particular, the authors emphasize that, in the coming years, the most important factors that will determine the success of controlling the COVID-19 pandemic will be the level of vaccination uptake by the population, and the effective use of resources to administer this. Moreover, they argue that game theory, supplemented by social network analysis and agent-based modelling, should be extensively used by researchers to model vaccination uptake by populations and guide difficult policy decisions regarding vaccination programs, and that due to limited ability effective prioritisation and optimal use of resources would be crucial.

This is in fact reality in many countries around Europe and the world right now, and I would thus find it important that this is published as soon as possible. The writing is clear and comprehensive, the arguments are convincing and sound, and there is in general clear that the work has benefit from ample experience and insight from a large number of authors with dedication and love to detail.

I agree that game theory, together with social networks and agent-based modeling, can and should be used as a primary theoretical framework in determining effective prioritisation of scarce resources needed in different vaccination programs, depending on a broad range of bounding conditions for successful implementation. To further support this fact, perhaps the authors can further improve the introduction by referring also to Imitation dynamics of vaccination behaviour on social networks, *Proceedings of the Royal Society B* 278, 42 (2011) and Imperfect Vaccine Aggravates the Long-Standing Dilemma of Voluntary Vaccination, *PloS one* 6, e20577 (2011), where this has been studied.

The conclusion could also do with a return to the key point, not just reiterating them, but also discussing possible/likely bottlenecks and challenges.

Other than that, as mentioned, I am warmly in favor of publication subject only to minor revision.

Decision letter (RSOS-210429.R0)

Dear Professor Perc

On behalf of the Editors, we are pleased to inform you that your Manuscript RSOS-210429 "Optimal governance and implementation of vaccination programs to contain the COVID-19 pandemic" has been accepted for publication in Royal Society Open Science subject to minor revision in accordance with the referees' reports. Please find the referees' comments along with any feedback from the Editors below my signature.

We invite you to respond to the comments and revise your manuscript. Below the referees' and Editors' comments (where applicable) we provide additional requirements. Final acceptance of

your manuscript is dependent on these requirements being met. We provide guidance below to help you prepare your revision.

Please submit your revised manuscript and required files (see below) no later than 7 days from today's (ie 12-May-2021) date. Note: the ScholarOne system will 'lock' if submission of the revision is attempted 7 or more days after the deadline. If you do not think you will be able to meet this deadline please contact the editorial office immediately.

on behalf of Dr Feng Fu (Associate Editor) and Mark Chaplain (Subject Editor)
openscience@royalsociety.org

Associate Editor Comments to Author (Dr Feng Fu):
Comments to the Author:

Reviewers uniamously recommend acceptance with minor revisions. Perhaps it is helpful to discuss the implications of vaccine effectiveness in both pathogen virulence evolution (e.g., Nature 414, no. 6865 (2001): 751-756, Nature Reviews Microbiology 18, no. 5 (2020): 265-265.) and uptake behavior (Scientific reports 7, no. 1 (2017): 1-9, Proceedings of the Royal Society B 286, no. 1894 (2019): 20182406). Look forward to receiving your revised manuscript.

Reviewer comments to Author:
Reviewer: 1

Comments to the Author(s)

In this work, authors study a very important and modern topic, i.e. strategies for containing SARS-COV-2 in the light of the production of vaccines, now available in some countries. Notably, they argue that the optimal strategy for distributing vaccines should result from an approach able to exploit both game theory and social network analysis.

The introduction reads very well, it is compact and presents the essential elements to the reader. The position statements are clear, as the authors clarify that their statements are based on a list of assumptions they explain. The background material is very well-written, also for non-expert in the field, providing all important definitions, as the value R_0 .

The motivation for using Game Theory is given in Section 2.3 and 4.1.

Then in 4.3 it is discussed how to couple this approach with simulations. Finally, Conclusion section provides a general overview of this contribution, highlighting motivations, methods, and

so explaining why this strategy should be actually considered/implemented by various governments.

In my opinion, as I above mentioned, this paper is very well-written and it covers a modern aspects in the current pandemic scenario. The discussion authors propose deserve full attention, not only from the scientific community. Their analysis is supported by clear statements and assumptions. For these reasons, I am glad to recommend this paper in the current form.

As optional comment, I would like to mention two investigations that authors could consider when preparing the final version of this work: 'An epidemiological model with voluntary quarantine strategies governed by evolutionary game dynamics, *Chaos, Solitons & Fractals* 143, 110616, 2021' where a model based on an Evolutionary Game is integrated in the dynamics of a SIR model, and 'Heterogeneity in evolutionary games: an analysis of the risk perception, *Proceedings of the Royal Society A* 476 (2237), 20200116, 2020'.

Notably, the first manuscript help to corroborate the the main claim proposed here, on the relevance of using game theory-based approach for dealing with COVID19 and similar scenarios. The second paper can be a useful reference for those readers interested in the impact of risk perception in dilemma games.

To conclude, I wish to congratulate with the authors for a such interesting and clear work.

Reviewer: 2

Comments to the Author(s)

In their paper authors study a critically relevant and important problem that can help better stem the burden of the COVID-19 pandemic with proper vaccination governance and implementation. In particular, the authors emphasize that, in the coming years, the most important factors that will determine the success of controlling the COVID-19 pandemic will be the level of vaccination uptake by the population, and the effective use of resources to administer this. Moreover, they argue that game theory, supplemented by social network analysis and agent-based modelling, should be extensively used by researchers to model vaccination uptake by populations and guide difficult policy decisions regarding vaccination programs, and that due to limited ability effective prioritisation and optimal use of resources would be crucial.

This is in fact reality in many countries around Europe and the world right now, and I would thus find if important that this is published as soon as possible. The writing is clear and comprehensive, the arguments are convincing and sound, and there is in general clear that the work has benefit from ample experience and insight from a large number of authors with dedication and love to detail.

I agree that game theory, together with social networks and agent-based modeling, can and should be used as a primary theoretical framework in determining effective prioritisation of scarce resources needed in different vaccination programs, depending on a broad range of bounding conditions for successful implementation. To further support this fact, perhaps the authors can further improve the introduction by referring also to Imitation dynamics of vaccination behaviour on social networks, *Proceedings of the Royal Society B* 278, 42 (2011) and Imperfect Vaccine Aggravates the Long-Standing Dilemma of Voluntary Vaccination, *PloS one* 6, e20577 (2011), where this has been studied.

The conclusion could also do with a return to the key point, not just reiterating them, but also discussing possible/likely bottlenecks and challenges.

Other than that, as mentioned, I am warmly in favor of publication subject only to minor revision.

===PREPARING YOUR MANUSCRIPT===

===PREPARING YOUR REVISION IN SCHOLARONE===

- An individual file of each figure (EPS or print-quality PDF preferred [either format should be produced directly from original creation package], or original software format).
 - An editable file of each table (.doc, .docx, .xls, .xlsx, or .csv).
 - An editable file of all figure and table captions.
- Note: you may upload the figure, table, and caption files in a single Zip folder.
- Any electronic supplementary material (ESM).
 - If you are requesting a discretionary waiver for the article processing charge, the waiver form must be included at this step.
 - If you are providing image files for potential cover images, please upload these at this step, and inform the editorial office you have done so. You must hold the copyright to any image provided.
 - A copy of your point-by-point response to referees and Editors. This will expedite the preparation of your proof.

- Ensure that your data access statement meets the requirements at <https://royalsociety.org/journals/authors/author-guidelines/#data>. You should ensure that you cite the dataset in your reference list. If you have deposited data etc in the Dryad repository, please only include the 'For publication' link at this stage. You should remove the 'For review' link.
- If you are requesting an article processing charge waiver, you must select the relevant waiver option (if requesting a discretionary waiver, the form should have been uploaded at Step 3 'File upload' above).
- If you have uploaded ESM files, please ensure you follow the guidance at <https://royalsociety.org/journals/authors/author-guidelines/#supplementary-material> to include a suitable title and informative caption. An example of appropriate titling and captioning may be found at https://figshare.com/articles/Table_S2_from_Is_there_a_trade-off_between_peak_performance_and_performance_breadth_across_temperatures_for_aerobic_scope_in_teleost_fishes_/3843624.

Author's Response to Decision Letter for (RSOS-210429.R0)

See Appendix A.

Decision letter (RSOS-210429.R1)

Dear Professor Perc,

I am pleased to inform you that your manuscript entitled "Optimal governance and implementation of vaccination programs to contain the COVID-19 pandemic" is now accepted for publication in Royal Society Open Science.

COVID-19 rapid publication process:

We are taking steps to expedite the publication of research relevant to the pandemic. If you wish, you can opt to have your paper published as soon as it is ready, rather than waiting for it to be published the scheduled Wednesday.

This means your paper will not be included in the weekly media round-up which the Society sends to journalists ahead of publication. However, it will still appear in the COVID-19 Publishing Collection which journalists will be directed to each week (<https://royalsocietypublishing.org/topic/special-collections/novel-coronavirus-outbreak>).

If you wish to have your paper considered for immediate publication, or to discuss further, please notify openscience_proofs@royalsociety.org and press@royalsociety.org when you respond to this email.

on behalf of Dr Feng Fu (Associate Editor) and Mark Chaplain (Subject Editor)
openscience@royalsociety.org

Appendix A

RESPONSE TO REVIEWER COMMENTS

Please note: As instructed, two MS Word files are being submitted, one with “highlights” where the changes are highlighted, and another which is a “clean” version without these highlights.

In the version with highlights, the changes made in response to reviewer and editorial comments are highlighted in **Maroon Bold**. We were also constrained to make some changes voluntarily, to reflect the latest developments in vaccination efforts worldwide since our first submission, and these voluntary changes are highlighted in **Purple bold**.

Editor:

Reviewers uniamously recommend acceptance with minor revisions. Perhaps it is helpful to discuss the implications of vaccine effectiveness in both pathogen virulence evolution (e.g., Nature 414, no. 6865 (2001): 751-756, Nature Reviews Microbiology 18, no. 5 (2020): 265-265.) and uptake behavior (Scientific reports 7, no. 1 (2017): 1-9, Proceedings of the Royal Society B 286, no. 1894 (2019): 20182406). Look forward to receiving your revised manuscript.

Response:

We thank the editor. We now briefly discuss the implications of vaccine effectiveness to both pathogen virulence evolution and uptake behaviour, and the recommended references have been cited.

Reviewer 1:

In this work, authors study a very important and modern topic, i.e. strategies for containing SARS-COV-2 in the light of the production of vaccines, now available in some countries. Notably, they argue that the optimal strategy for distributing vaccines should result from an approach able to exploit both game theory and social network analysis.

The introduction reads very well, it is compact and presents the essential elements to the reader. The position statements are clear, as the authors clarify that their statements are based on a list of assumptions they explain. The background material is very well-written, also for non-expert in the field, providing all important definitions, as the value R_0 .

Response: *We thank the reviewer for these comments.*

The motivation for using Game Theory is given in Section 2.3 and 4.1.

Then in 4.3 it is discussed how to couple this approach with simulations. Finally, Conclusion section provides a general overview of this contribution, highlighting motivations, methods, and so explaining why this strategy should be actually considered/implemented by various governments.

In my opinion, as I above mentioned, this paper is very well-written and it covers a modern aspects in the current pandemic scenario. The discussion authors propose deserve full attention, not only from the scientific community. Their analysis is supported by clear statements and assumptions. For these reasons, I am glad to recommend this paper in the current form.

Response: *We thank the reviewer for these comments.*

As optional comment, I would like to mention two investigations that authors could consider when preparing the final version of this work: 'An epidemiological model with voluntary quarantine strategies governed by evolutionary game dynamics, Chaos, Solitons & Fractals 143, 110616, 2021' where a model based on an Evolutionary Game is integrated in the dynamics of a SIR model, and "Heterogeneity in evolutionary games: an analysis of the risk perception, Proceedings of the Royal Society A 476 (2237), 20200116, 2020".

Notably, the first manuscript help to corroborate the the main claim proposed here, on the relevance of using game theory-based approach for dealing with COVID19 and similar scenarios. The second paper can be a useful reference for those readers interested in the impact of risk perception in dilemma games.

Response: *We have included these references and very briefly discussed their content in the final version.*

To conclude, I wish to congratulate with the authors for a such interesting and clear work.

Response: *We thank the reviewer for these comments.*

Reviewer 2:

In their paper authors study a critically relevant and important problem that can help better stem the burden of the COVID-19 pandemic with proper vaccination governance and implementation. In particular, the authors emphasize that, in the coming years, the most important factors that will determine the success of controlling the COVID-19 pandemic will be

the level of vaccination uptake by the population, and the effective use of resources to administer this. Moreover, they argue that game theory, supplemented by social network analysis and agent-based modelling, should be extensively used by researchers to model vaccination uptake by populations and guide difficult policy decisions regarding vaccination programs, and that due to limited ability effective prioritisation and optimal use of resources would be crucial.

This is in fact reality in many countries around Europe and the world right now, and I would thus find it important that this is published as soon as possible. The writing is clear and comprehensive, the arguments are convincing and sound, and there is in general clear that the work has benefit from ample experience and insight from a large number of authors with dedication and love to detail.

Response: *We thank the reviewer for these comments.*

I agree that game theory, together with social networks and agent-based modeling, can and should be used as a primary theoretical framework in determining effective prioritisation of scarce resources needed in different vaccination programs, depending on a broad range of bounding conditions for successful implementation.

To further support this fact, perhaps the authors can further improve the introduction by referring also to Imitation dynamics of vaccination behaviour on social networks, Proceedings of the Royal Society B 278, 42 (2011) and Imperfect Vaccine Aggravates the Long-Standing Dilemma of Voluntary Vaccination, PloS one 6, e20577 (2011), where this has been studied.

Response: *We have now cited the abovementioned papers in the introduction or background, and briefly discussed them.*

The conclusion could also do with a return to the key point, not just reiterating them, but also discussing possible/likely bottlenecks and challenges.

Response: *we have now re-iterated in the conclusion the key points the paper makes, and briefly discussed some bottlenecks and challenges in this approach.*

Other than that, as mentioned, I am warmly in favor of publication subject only to minor revision.

Response: *again, we thank the reviewer.*